# Test Strategy Optimization Based on Soft Sensing and Ensemble Belief Measurement

**DOI:** 10.3390/s22062138

**Published:** 2022-03-10

**Authors:** Wenjuan Mei, Zhen Liu, Lei Tang, Yuanzhang Su

**Affiliations:** 1School of Automation Engineering, University of Electronic Science and Technology of China, Chengdu 611731, China; meiwenjuan@std.uestc.edu.cn (W.M.); syz@uestc.edu.cn (Y.S.); 2Southwest Institute of Technical Physics, Chengdu 611731, China; ltang20001142@163.com; 3School of Foreign Language, University of Electronic Science and Technology of China, Chengdu 611731, China

**Keywords:** prognostic and health management, extreme learning machine, soft sensors

## Abstract

Resulting from the short production cycle and rapid design technology development, traditional prognostic and health management (PHM) approaches become impractical and fail to match the requirement of systems with structural and functional complexity. Among all PHM designs, testability design and maintainability design face critical difficulties. First, testability design requires much labor and knowledge preparation, and wastes the sensor recording information. Second, maintainability design suffers bad influences by improper testability design. We proposed a test strategy optimization based on soft-sensing and ensemble belief measurements to overcome these problems. Instead of serial PHM design, the proposed method constructs a closed loop between testability and maintenance to generate an adaptive fault diagnostic tree with soft-sensor nodes. The diagnostic tree generated ensures high efficiency and flexibility, taking advantage of extreme learning machine (ELM) and affinity propagation (AP). The experiment results show that our method receives the highest performance with state-of-art methods. Additionally, the proposed method enlarges the diagnostic flexibility and saves much human labor on testability design.

## 1. Introduction

With the increasing use of electric devices, prognostic and health management engineering (PHM engineering) has played an extremely significant role in product lifetime management over decades [1]. PHM engineering ensures electric devices’ lifetime healthy operation and provides appropriate resource assignment for product management [2]. In recent years, the production cycle has shortened because circuit technology and system design have rapidly developed [3,4,5,6]. The system structures have become more complicated, more integrated, more intelligent, and highly intensive [7]. Additionally, the potential test procedures and fault cases grow exponentially. As a result, PHM engineering has received active demand and new challenges. Practical conditional maintenance (CM) solutions become difficult to generate on modern system applications. On another hand, CM must be flexible enough to math structure complexity and system function complexity. Hence, efficient PHM engineering solutions for modern devices become an urgent problem for academic researchers and industrial engineers.

Under the CM design, testability design and maintainability design are two essential projects to determine supportability, enhance reliability, and guarantee safety during lifetime device management [8]. Testability design analyses the system’s internal structures, selects test projects and arranges test procedures, estimates the system operation condition, and locates failure modes. The key difficulty for testability design is balancing the system’s high structural complexity and the solution efficiency properly. Classical testability design approaches use dynamic programming (DP) to assess optimal solutions [9,10]. However, classic methods suffer from high time complexity when either the number of test projects or failures is larger than 12.

In recent years, the AO* method [11], a sequential testing generation method, balances the generation complexity of test solutions and detection performance for diagnostic procedures with heuristic searching and AND/OR graph topology. Thus, the AO* method became the most popular testability design technique. To match the growing electric system complexity, AO* has been improved to optimize searching mechanisms with information theory [12], evolution algorithms [13,14], dynamic design [15], etc. Additionally, advanced research has simplified searching procedures with rollout strategy [16] and bottom-up decision tree [17] to achieve practical large-system testability solutions. Despite good application results, these testability design approaches need to assign logic relationships between test procedures and failure modes. Hence, all these methods assume that dependent single-signal operations and sequential procedures can make all diagnostic decisions. However, modern devices contain highly complicated logistic relationships [18] between test procedures and potential failure modes under many scenarios. Consequently, the testability design consumes too much human power and affects the testability design efficiency to prepare prior knowledge, especially under short production cycles. Furthermore, the testability design wastes the entailed sensor recording information from test procedures as the existing methods rely on human-selected binary information.

The maintainability estimation model provides real-time operation condition diagnosis and realizes system health management along with testability design. In general, existing maintainability design approaches can be divided into physics-of-failure (PoF) approaches [19,20,21,22] and data-driven (DD) approaches [23,24,25]. PoF approaches use rules from physics or chemical dynamics to estimate electric system failure conditions [26]. With accelerated aging of experimental records and prior modeling knowledge, PoF approaches generate an accurate dynamic model under specific stress influences such as thermal, electrical, and humidity. However, most PoF approaches are only suitable under one stress function, and the methods face high limitations with real applications.

Unlike PoF methods, DD approaches rely on historical sensor information and build maps from sensor recording to failure modes. Hence, DD approaches become more flexible compared to PoF methods. Classical DD approaches use statistic methods such as stochastic methods, regression methods, distance estimation, and similarity estimation [27,28,29,30]. These methods require large samples to ensure unbiased estimation and model robustness, and are constrained in off-line modeling. Therefore, statistical approaches have limitations on maintainability design with few sampling records and short time constraints. In recent years, machine learning (ML) methods [31,32] have attracted research attention because of its high accuracy, strong adaptive ability, powerful robustness, and fast computation time. As a result, ML methods, such as neural networks (NN) [33], support vector machine (SVM) [34,35], k-nearest neighbors (KNN) [36], K-means clustering [37], and particle filters (PF) [38], have been used for maintenance design and tuned to successfully extract hidden rules from failure mode and recording information [39]. However, these methods require prior information selection and the preprocessing system degradation features. The performance of testability and maintainability may affect the preprocessing model input due to the complicated system structure and the complex recording information relationships. 

We propose a test strategy optimization based on this paper’s soft-sensing and ensemble belief measurement to overcome this weakness described above. We suggest a closed loop framework for PHM design to replace the sequential framework between testability and maintenance design. Instead of experienced knowledge, our method uses ensemble learning based on direct sensor recordings to gain the system state estimation. Additionally, we connect the testability and maintenance design by the minimal conditional criterion to optimize the test strategy. Consequently, the proposed method improves the flexibility of PHM design, saves labor, and enhances diagnostic efficiency. The contributions of our work follow.

Our model builds diagnostic strategies without much prior knowledge and human-elected features. The diagnostic tree is constructed with ELM-based soft-sensor nodes. Instead of experienced features, ELM-based soft-sensor nodes provide basic probability assignment (BPA) directly from the sensor records. Hence, our methods cut the testability design human labor since the method needs no system mechanisms analysis.We build a closed loop between testability design and maintenance design. Thus, the maintenance design makes full use of testability design information and improves the testability design efficiency with the advantage of ELM-based construction modules.Our model divides the fault set adaptively into several fuzzy sets with affinity propagation and improves the diagnostic efficiency of single test procedures.

The experiment proves that our method has better diagnostic accuracy and lower false alarm ratios than other state-of-art diagnostic methods. Additionally, our diagnostic strategies take only a few tests with little test assignment consumption. For each fault state, the diagnostic procedures provide one efficient test sequence. Thus, the diagnostic procedure enjoys high efficiency for applications. Finally, affinity propagation enlarges the diagnostic flexibility and significantly reduces human labor used on testability design.

The rest of this paper is organized as follows. We introduce the PHM design problem and provide the general framework in the next section. Section 3 presents the details of the algorithms and Section 4 provides the experiment results and discussions. Finally, conclusions are drawn in Section 5.

## 2. Problem Formulation and General Framework

PHM engineering estimates the degradation processes and recognizes failure modes over the product’s full lifetime. Based on experimental research and application surveys, PHM engineering provides fault analysis and maintenance advice to prevent failure occurrence. PHM engineering elements and the corresponding relationships are presented in Figure 1.

To study the target systems, engineers analyze potential failure modes and find unsafe and unreliable features. Thus, the target system’s safety and reliability, reflected by its failure features, draw attention to system failure physical characteristics. For higher reliability and safety, the PHM platform provides maintenance procedures and diagnostic approaches for the system’s supportability. The diagnostic approach depends on testability and the maintenance procedures that rely on maintainability. 

Testability is the design characteristic that the health condition can be detected accurately and failure modes can be located successfully. Meanwhile, maintainability is the ability to repair and recover the system under certain conditions within a certain time. The PHM services can ensure healthy system operation and achieve the PHM engineering purpose with proper maintenance and testability design. In general, supportability is the key PHM engineering purpose influenced by testability and maintainability; thus, the PHM must meet the environmental stability needs. The physical system structure directly influences safety and reliability while system elements reflect safety and reliability critically. High testability and maintainability quality improve the system reliability and safety with good maintenance design procedures and diagnostic approaches. Hence, testability design and maintenance design are two significant PHM platform parts.

Since testability and maintenance are important, various techniques provide practical testability design and maintenance design. To our best knowledge, existing methods build a sequential approach to generate the PHM service. As Figure 2 shows, the traditional framework arranges the test procedures to assess the sensor recording and create maintenance design with the assigned sensor records. The framework is applicable for many large systems. However, the testability design uses human experience binary information such as information flow chart, dependence matrix and AND/OR graph. Therefore, the testability design suffers from extended time and the financial burden with modern complex systems, ignores the coupling effects between test procedures, and wastes valuable sensor recording information. As a result, the maintenance design receives poor performance and low efficiency because of poor information usage and low knowledge transmission efficiency from testability design and maintenance design.

We introduce a closed-loop test strategy optimization method on soft-sensor information and ensemble learning to overcome the weakness. Similar to traditional PHM design approaches, the proposed method aims to generate a fault diagnostic tree and cut the fault set with testability sensor information and maintenance signal processing. In contrast, our fault tree grows with direct sensor recording information directly along with the processing module and extends with basic probability assignments. On the other hand, the PHM design process contains a cooperative closed loop between testability and maintenance design to improve information usage and transmission efficiency during fault detection.

The general framework of the proposed method is presented in Figure 3. For maintenance design, the soft sensor integrates the recording information from assigned sensors and the processing signal, such as statistical method, support machine, and learning machine to extract suitable test features. Considering the requirement of fast diagnosis and detection, we use extreme learning machine (ELM) [34], a noniterative single-layer learning machine, to generate accurate fast processing modules of soft-sensing nodes. The details are introduced in the next section.

Suppose there exists *N* possible sensors with the PHM system design; therefore, the potential test procedures set is denoted as $T_{potential}=\left\{t_{1},t_{2},\ldots ,t_{i},\ldots t_{N-1},t_{N}\right\}$, where $t_{i}$ means the test procedures with *i*-th sensors. As each sensor contains a vector of information $x_{i}=\left\{x_{i,1},x_{i,2},\ldots ,x_{i,K}\right\},\;i=1,2,3,\ldots ,N$ with *M* existing samples from the targeted system, the training information is regarded as:(1)$$X_{train}=\left[\begin{array}{cccc} x_{1,1} & x_{1,2} & \ldots & x_{1,N}\\ x_{2,1} & x_{2,2} & \ldots & x_{2,N}\\ \vdots & \vdots & \ddots & \vdots \\ x_{M,1} & x_{M,2} & \ldots & x_{M,N} \end{array}\right]$$
where $x_{ij}$ is the recording information vector from the *j*-th sensor for *i*-th sample, *j* = 1,2,3,…, *N*, and *i* = 1,2,3,…, *M*.

With the sampling information, the potential fault tree soft-sensor node structure is denoted as following:(2)$$Node=\left\{S_{father},S_{son},T_{node},X_{node},Y_{node},ELM_{node},m_{exemplar}\right\}$$
where the $T_{node}$ is the assigned test procedures from previous selected soft-sensor nodes and the potential selected test procedure, and $X_{node}$ is the training sample sensor recording information to detect under the node, as follows:(3)$$X_{node}={\left[x_{1,T_{node}},x_{2,T_{node}},\ldots ,x_{M_{node},T_{node}}\right]}^{T}$$
(4)$$x_{i,T_{node}}=\{x_{i,j}|t_{j}\in T_{node}\}$$
where $M_{mode}$ is the number of the sampling data. $Y_{node}$ provides the corresponding sample failure conditions. Suppose the whole set of failure mode is $S_{node}=\left\{s_{1},s_{2},\ldots ,s_{K}\right\}$, where *K* is the number of fault modes considered, then $Y_{node}$ is determined with the actual condition of training samples as follows:(5)$$Y_{node}={\left[y_{1},y_{2},\ldots ,y_{M_{node}}\right]}^{T}$$
(6)$$y_{i}=\left[y_{i,1},y_{i,2},\ldots ,y_{i,K}\right]$$
(7)$$y_{i,j}=\{\begin{array}{cc} 1 & ss_{i}=s_{j}\\ 0 & ss_{i}\neq s_{j} \end{array}$$
where $ss_{i}$ is the actual *i*-th sample fault mode. Additionally, $S_{father}$ is the fault mode set of the soft-sensor node, represented as follows:(8)$$S_{father}=\{s_{j}|\exists x_{i}\in X_{node}\;and\;s_{j}\in S,\;ss_{i}=s_{j}\}$$

With $X_{node}$ and $Y_{node}$, the ELM serves as the soft-sensor signal processing part and aims to provide the fuzzy set of fault states as much as possible. To achieve the goal, ELM builds a map $f:R^{1\times N}\to R^{1\times K}$ from the training sample signals to estimate the fault states and provides the training sample prediction $M_{node}$ as follows:(9)$$M_{node}={\left[f(x_{1,T_{node}}|ELM_{node}),f(x_{2,T_{node}}|ELM_{node}),\ldots ,f(x_{M_{node},T_{node}}|ELM_{node})\right]}^{T}$$

With $M_{node}$, ELM estimates the training sample failure modes. To determine the processing part performance, $Y_{node}$ is used as the expected output marks for the fault detection process. From the view of detection process, two indexes, fault detection rate (FDR) and false alarm rate (FAR), play essential roles in evaluating the accuracy.

FDR is defined as the ratio between the failure mode probability that is successfully detected with the ELM and the total failure modes probability. Here, we assume the historical samples are subject to the general failure probability distribution of the real applications. Thus, the statistic characteristics of training samples reflect the total failure mode probability and the training sample detection performance depicts the detection probability. From above, FDR for the node is presented as follows:
(10)$$FDR_{node}=\frac{\sum_{i=1}^{M_{node}}\sum_{j=1}^{K}P(f_{j}(x_{1,T_{node}}|ELM_{node})\geq 1-\varepsilon ,y_{i,j}=1)}{\sum_{i=1}^{M_{node}}\sum_{j=1}^{K}y_{i,j}}$$
where ε is the detection margin. From the generation process, $f_{j}(x_{1,T_{node}}|ELM_{node})$ and $y_{i,j}$ are independent of each other. Additionally, since ELM provides a continuous probability estimation, the loss function with respect to $FDR_{node}$ is computed as follows:(11)$$L_{FDR_{node}}=\frac{\sum_{i=1}^{M_{node}}\sum_{j=1}^{K}\left(y_{i,j}-f_{j}\left(x_{1,T_{node}}|ELM_{node}\right)\right)P\left(y_{i,j}=1\right)}{\sum_{i=1}^{M_{node}}\sum_{j=1}^{K}y_{i,j}}$$

Along with FDR, FAR presents the ratio between false-alarm failure mode probability and the total failure detection probability. Taking consideration of $f_{j}(x_{1,T_{node}}|ELM_{node})$, FAR is computed as follows:(12)$$FAR_{node}=\frac{\sum_{i=1}^{M_{node}}\sum_{j=1}^{K}P\left(f_{j}\left(x_{1,T_{node}}|ELM_{node}\right)\geq 1-\varepsilon ,y_{i,j}=0\right)}{\sum_{i=1}^{M_{node}}\sum_{j=1}^{K}P\left(f_{j}\left(x_{1,T_{node}}|ELM_{node}\right)\geq 1-\varepsilon \right)}$$

Here, it is assumed that the model has enough accuracy so that the failure estimation and the total failure conditions have approximate values. Thus, $\sum_{i=1}^{M_{node}}\sum_{j=1}^{K}P\left(f_{j}\left(x_{1,T_{node}}|ELM_{node}\right)\geq 1-\varepsilon \right)$ is able to be approximately equal to $\sum_{i=1}^{M_{node}}\sum_{j=1}^{K}y_{i,j}$. Therefore, the loss function of the FAR is denoted as:(13)$$\begin{array}{ll} L_{node} & =\frac{\sum_{i=1}^{M_{node}}\sum_{j}^{K}{\left(y_{i,j}-f_{j}\left(x_{1,T_{node}}|ELM_{node}\right)\right)}^{2}P\left(y_{i,j}=1\right)}{\sum_{i=1}^{M_{node}}\sum_{j}^{K}y_{i,j}}\\ & \;\;\;\;\;+\frac{\sum_{i=1}^{M_{node}}\sum_{j}^{K}{\left(y_{i,j}-f_{j}\left(x_{1,T_{node}}|ELM_{node}\right)\right)}^{2}P(y_{i,j}=0)}{\sum_{i=1}^{M_{node}}\sum_{j}^{K}y_{i,j}}\\ & \;\;\;\;\;=\frac{\sum_{i=1}^{M_{node}}\sum_{j}^{K}{\left(y_{i,j}-f_{j}\left(x_{1,T_{node}}|ELM_{node}\right)\right)}^{2}}{\sum_{i=1}^{M_{node}}\sum_{j}^{K}y_{i,j}} \end{array}$$

As $\sum_{i=1}^{M_{node}}\sum_{j}^{K}y_{i,j}$ is independent from the soft-sensor construction, the task of ELM is to minimize $\sum_{i=1}^{M_{node}}\sum_{j}^{K}{\left(y_{i,j}-f_{j}\left(x_{1,T_{node}}|ELM_{node}\right)\right)}^{2}$, which is the difference between $M_{node}$ and $Y_{node}$, expressed as follows:(14)$$ELM=argmin\left\{\sum_{i=1}^{M_{node}}||y_{i}-f(x_{1,T_{node}}|ELM_{node})|{|}^{2}\right\}$$

Since the maintenance design of the proposed method directly relies on the recording sensor signals, the physical system knowledge is largely preserved and the information usage is highly enhanced. 

Based on soft-sensor node construction, testability design process adds the soft-sensor node with best performance and builds a fault tree, taking consideration of potential soft-sensor node under the minimum conditional entropy criterion. Hence, the assigned soft-sensor nodes decrease the diagnostic uncertainty and improve the detection efficiency. Besides, affinity propagation (AP) is adapted to separate the fuzzy set of the fault modes and generate subnodes for the diagnostic model with the exemplar probability estimation $m_{exampor}$ and basic probability assignment $BPA_{node}$. The subnodes are denoted with the subset of failure modes $S_{son}=\left[S_{son,1},S_{son,2},\ldots ,S_{son,K_{node}}\right]$ and satisfies the condition that $\cup_{i=1}^{K_{node}}S_{son,i}=S_{father}$.

After adding the soft-sensor nodes and extending the subset of fault modes, the information of assigned nodes $T_{node}$ is regarded as prior knowledge, serving as feedback from testability design to maintenance design, and extending the fault tree until reaching the minimum fault condition set.

With the cooperation between testability design and maintenance design, a PHM model based on soft-senor information is generated and the sensors for PHM maintenance are assigned based on the selected test set of PHM model, as follows.
(15)$$T_{PHM}=\left\{t_{i}:\exists T_{node},\;t_{i}\in T_{node}\right\}$$

To locate the fault condition when starting with the maximum set of failure modes, the corresponding sensor recording is collected and used to compute the potential basic probability assignment. Then, the subset of potential modes is determined based on existing samples with nearest neighbor strategy and the detection process is continued until finding the minimum failure sets and obtaining the failure detection. For each test sample denoted as $case_{i}$, the PHM detection procedures generate a test sequence corresponding to the assigned sensor recording and the estimation from the signal processing by a branch of assigned soft-sensor nodes, as follows:(16)$$S_{node}=\left\{Node_{j}:\exists m_{exemplar,k}\in m_{exemplar},\left|\left|f\left(x_{case_{i},T_{node,j}}|ELM_{node_{j}}\right)-m_{exemplar,k}\right|\right|=d_{node}\right\}$$
where $d_{Node_{j}}$ is the minimum distance from the failure condition estimation vector $f(x_{case,T_{node}}|ELM_{node_{j}})$ to all exemplars of soft-sensor nodes with the same father node of $Node_{j}$. The detection process ends when the procedures reach the terminal node of $S_{node}$ denoted as $Node_{terminal,case}$. Then, the estimated failure condition vector is computed as:(17)$${\hat{y}}_{case.j}=\{\begin{array}{c} \begin{array}{cc} 1 & s_{j}\in S_{node_{terminal,case},father} \end{array}\\ \begin{array}{cc} 0 & s_{j}\notin S_{node_{terminal,case},father} \end{array} \end{array}$$

According to the definitions of FDR, FAR, and detection accuracy, the test performance indexes of the detection procedures are computed as follows:(18)$$FDR_{test}=\sum_{X_{case}}\sum_{j=1}^{K}P({\hat{y}}_{case,j}=1|y_{casse,j}=1)$$
(19)$$FDR_{test}=\sum_{X_{case}}\sum_{j=1}^{K}P({\hat{y}}_{case,j}=0|y_{casse,j}=1)$$
(20)$$Accuracy_{test}=\sum_{X_{case}}\sum_{j=1}^{K}P\left({\hat{y}}_{case,j}=1|y_{casse,j}=1\right)+P({\hat{y}}_{case,j}=0|y_{casse,j}=0)$$

From these, the test strategy optimization aims to select $T_{PHM}$ from $T_{potential}$ and generate the diagnostic tree with soft sensors and AP. For each case, AP determines the next procedure and the corresponding soft sensors with previous estimations. Thus, for each case, the diagnostic tree provides an adaptive test sequence and leads to the final evaluation on the terminal node. Similar to Equation (4), the objective function is the combination of FAR loss and FDR loss, as follows:(21)$$L_{tree}=\frac{\sum_{i=1}^{M}\sum_{j=1}^{K}{\left(y_{i,j}-F_{j}(x_{i,T_{seq},i}|T_{seq,i}\in T_{PHM},ELM_{node}\in tree)\right)}^{2}}{\sum_{i=1}^{M}\sum_{j=1}^{K}y_{i,j}}$$
where $T_{seq,i}$ is the required test procedure with respect to the diagnostic tree, and $F_{j}(x_{i,T_{seq},i}|T_{seq,i}\in T_{PHM},ELM_{node}\in tree)$ is the state estimation from the terminal node with respect to the test procedure of the *i*-th case.

## 3. Test Strategy Optimization Based on Soft Sensing and Ensemble Belief Measurement

### 3.1. Construct Soft-Sensor Node with Extreme Learning Machine

As mentioned in the previous section, each soft sensor contains the recording information from the assigned sensors, the artificial intelligence signal processing modules, and probability estimation parameter for the isolation of fault states. During the construct process, maintenance design produces the soft-sensor node with candidate test procedures and candidate soft-sensor nodes with high performance, and generates the fault tree. For each candidate node, the sensor recording input is created as follows:(22)$$X_{candidate,\;node}=\left\{x_{s_{i},T_{i}^{*}}:x_{i}\in X_{candidate\;node}\right\}$$
(23)$$x_{s_{i},T_{i}^{*}}=\{x_{s_{i},t}|t\in T_{i}^{*}\}$$
(24)$$T_{i}^{*}=\left\{T_{sequence}^{*},t^{*}\right\}\;\mathrm{where}\;\left\{t^{*}\right\}\cap T_{sequence}^{*}=\emptyset \;$$
where $T_{sequence}^{*}$ integrates the previous test information of before the candidate node and makes full use of sensor recording knowledge.

At the same time, we use ELM to generate artificial intelligence signal processing modules for fast training and high generation ability. Shown in Figure 4, ELM is a noniterative three-layer neural network and contains parameters of a fully connected hidden layer and a linear-combined output layer with an activation function, as follows:(25)$$ELM_{candidate,node}=\left\{\left(W_{node},b_{node}\right),\beta_{\mathrm{node}},f_{h}\left(.\right)\right\}$$
(26)$$W_{node}=\left[\begin{array}{ccc} w_{1,1} & \ldots & w_{1,N_{T}}\\ w_{2,1} & \ldots & w_{2,N_{T}}\\ \vdots & \ddots & \vdots \\ w_{L,1} & \ldots & w_{L,N_{T}} \end{array}\right]$$
(27)$$b_{node}=\left[b_{node.1},b_{node.2},\ldots ,b_{node.L}\right]$$
(28)$$\beta_{node}=\left[\begin{array}{ccc} \beta_{1,1} & \ldots & \beta_{1,N_{T}}\\ \beta_{2,1} & \ldots & \beta_{2,N_{T}}\\ \vdots & \ddots & \vdots \\ \beta_{L,1} & \ldots & \beta_{L,N_{T}} \end{array}\right]$$
where $W_{node}$ is denoted as the weights matrix of the hidden layer while $b_{node}$ is the hidden layer bias, and *L* is the number of hidden nodes. $\beta_{node}$ is the output layer weight and $f_{h}\left(.\right)$ determines the activation function from the sensor input and hidden output. Here, the sigmoid function is taken as the activation functions for all soft-sensor nodes. Relative to $X_{candidate\;node}$, the hidden outputs of training samples are produced as follows:(29)$$H_{node}=\left[\begin{array}{ccc} f_{h}\left(w_{1}x_{1,T^{*}}+b_{node,1}\right) & \ldots & f_{h}\left(w_{L}x_{1,T^{*}}+b_{node,L}\right)\\ f_{h}\left(w_{1}x_{2,T^{*}}+b_{node,L}\right) & \ldots & f_{h}\left(w_{L}x_{2,T^{*}}+b_{node,L}\right)\\ \vdots & \ddots & \vdots \\ f_{h}\left(w_{1}x_{M_{node},T^{*}}+b_{node,1}\right) & \ldots & f_{h}\left(w_{L}x_{M_{node},T^{*}}+b_{node,L}\right) \end{array}\right]$$

As ELM is a noniterative learning machine, $W_{node}$ and $b_{node}$ can be assigned randomly with respect to arbitrary probability distribution, and the output of the model is computed as a linear combination of the hidden output with trained $\beta_{node}$**,** as follows:(30)$${\dot{Y}}_{candidate\;node}=\beta_{node}H_{node}$$

To estimate the failure situation as accurately as possible, ${\dot{Y}}_{candidate\;node}$ is supposed to be consistent with actual failure states $Y_{node}$ defined based on Equations (5)–(7). According to Equation (14), the loss function of the candidate node is computed as follows:(31)$$Loss_{candidate\;node}={\left|\left|Y_{node}-{\dot{Y}}_{candidate\;node}\right|\right|}^{2}={\left|\left|Y_{node}-\beta_{node}H_{node}\right|\right|}^{2}\;$$

Taking differential of $Loss_{candidate\;node}$ with respect to $\beta_{node}$, the trained output weight is accessed as follows:(32)$$\beta_{node}={\left(H_{node}H_{node}^{T}+\lambda I\right)}^{-1}H_{node}Y_{node}$$

Based on the proper assignment of ELM parameters, the soft-sensor nodes gain knowledge from the training samples and obtain accurate condition estimation with the test samples.

From above, considering the candidate node with sensor recording inputs $X_{candidate,\;node}$, the previous test sequence $T_{sequence}^{*}$, and candidate test point $t^{*}$, the procedure to generate the $ELM_{node}$ follows:


Step 1: Assign the candidate node with Equation (2), where $S_{father}=\{s_{i}|s_{candidate\;node}=s_{i},s_{i}\in S\}$. Meanwhile, $T_{node}$ is generated as Equation (24), $Y_{node}$ is assigned with Equations (5)–(7);Step 2: Initialize ELM parameters $\left(W_{node},b_{node}\right)$ randomly in [−1,1];Step 3: Calculate the hidden output with respect to $X_{candidate,\;node}$ as in Equation (29);Step 4: Train the output weights $\beta_{node}$ with Equation (32);Step 5: Obtain the estimation of candidate set ${\dot{Y}}_{candidate\;node}$ with Equation (30).


### 3.2. Separate the Fault Set Based on Affinity Propagation

With ELM-based soft-sensor nodes, the condition of trained samples and test samples can be estimated with high efficiency. Meanwhile, owing to the individual sensor recording knowledge limitation, the ELM condition evaluation has a vague part with unrelated failure modes. Thus, the fault set of corresponding nodes $S_{father}$ is divided into several fuzzy sets $S_{son}=\left[S_{son,1},S_{son,2},\ldots ,S_{son,K_{node}}\right]$ based on the fault state evaluation value ${\dot{Y}}_{candidate\;node}$. When constructing traditional diagnostic tree and fault analysis processes, the failure mode subset is divided by comparing the fault state evaluation and reference value of the failure mode or the failure mode calibration value. However, these strategies are only applicable to systems with small structures or systems with known mechanisms and a historical sample may ignore the diversity and validity. Hence, in the proposed method, we introduce a new dividing strategy based on affinity propagation (AP) to cut the fuzzy set and samples with evaluation similarity measurement between pairs of data points. With AP, $S_{son}$ it is constituted based on the similarity between condition estimates and the fault tree generation flexibility is enhanced.

Instead of assigning engineering-experience reference information, AP generates clusters based on all training set evaluation values ${\dot{Y}}_{candidate\;node}=\left\{y_{1},y_{2},\ldots ,y_{M_{node}}\right\}$. The clustering method treats each training sample as one data edge point and transmits two real-valued messages: the responsibility value $r\left(y_{i},y_{j}\right)$ and the availability value $a\left(y_{i},y_{j}\right)$, to realize communication between edge nodes until a good set of exemplars and corresponding clusters emerges. The responsibility value $r\left(y_{i},y_{j}\right)$ indicates the accumulated evidence for how well suited a historical data point $y_{j}$ is to serve as the exemplar for the historical data point $y_{i}$. In addition, the availability value $a\left(y_{i},y_{j}\right)$ represents how appropriate it would be for a historical data point $y_{i}$ to choose a historical data point $y_{i}$ as its exemplar. For initialization, the similarity between the historical points $\left(y_{i},y_{j}\right)$ is calculated based on Euclidean distance, as follows:(33)$$d\left(y_{i},y_{j}\right)=-{\left|\left|y_{i}-y_{j}\right|\right|}^{2}$$
where $d\left(y_{i},y_{j}\right)$ reflects how well the historical data point $y_{j}$ is suited to be the exemplar for historical data point $y_{i}$. AP aims to provide a clustering solution that satisfies the historical data points with larger values of distance estimation, which are more likely to serve as exemplars. To achieve this purpose, the clustering method recursively conducts the following updating process, sending the responsibility message from each data point to the corresponding data point.

First, the responsibility value $r\left(y_{i},y_{j}\right)$ is computed based on the following data driven approach:(34)$$r\left(y_{i},y_{j}\right)=d\left(y_{i},y_{j}\right)-\underset{k,s.t.k\neq j}{\max}\left\{a\left(y_{i},y_{k}\right)+d\left(y_{i},y_{k}\right)\right\}$$

As the availability value $a\;\left(y_{i},y_{j}\right)$ is set to 0, the responsibility value $r\left(y_{i},y_{j}\right)$ is initialized as the input similarity $d\left(y_{i},y_{j}\right)$ minus the largest similarity value between $y_{i}$ and other exemplars. Hence, the updating process does not consider how many other points favor each candidate exemplar. In later process, if some point is efficient to assign with other exemplars, the corresponding availability value will drop to less than 0 with the updating of $a\left(y_{i},y_{j}\right)$. Then, negative $a\left(y_{i},y_{j}\right)$ will decrease the effective value of the similarities value $d\left(y_{i},y_{j}\right)$ by Equation (34) and removes the corresponding candidate exemplars from competition. Especially, the self-responsibility value $r\left(y_{i},y_{j}\right)$ is set to the input preference that the training data point $y_{i}$ becomes one of the exemplars of clusters and reflects accumulated evidence that $y_{i}$ is an exemplar based on its input preference tempered by how ill-suited it is for assignment to another exemplar.

After calculating the responsibility value, the availability value is updated to gather evidence from the training data point as to whether each candidate exemplar makes a good exemplar, as follows:(35)$$a\left(y_{i},y_{j}\right)=min\left\{0,d\left(y_{i},y_{j}\right)+\underset{k,s.t.k\neq j}{\max}\left\{a\left(y_{i},y_{k}\right)+d\left(y_{i},y_{k}\right)\right\}\right\}$$

In addition, the self-availability value $a\left(y_{i},y_{j}\right)$ is updated as follows:(36)$$a\left(y_{i},y_{j}\right)=\sum_{k,s.t.k\neq j,k\neq i}max\left\{0,r\left(y_{i},y_{k}\right)\right\}$$
where $a\left(y_{i},y_{j}\right)$ reflects accumulated evidence that the training data point $y_{i}$ becomes an exemplar. For data point $y_{i}$, the data point $y_{i}$ that maximizes $a\left(y_{i},y_{j}\right)+r\left(y_{i},y_{j}\right)$ is chosen to be the exemplar. Additionally, if i=j, then it is necessary to identify the data point $y_{i}$ as the exemplar and assign its estimation value $y_{i}$ as the exemplar value $m_{exemplar}$. The set of all the $m_{exemplar}$ is denoted as $M_{exemplar}$. Based on each exemplar, one subset of $S_{father}$ satisfies.
(37)$$\begin{array}{ll} \forall m_{exemplar}\in & M_{exemplar},\exists S_{son,k}\subset S_{son,k}\;s.t.\;S_{son,k}\\ & \;\;\;=\{s_{i}:\exists x_{j}\in X_{node},\;a\left(y_{i},m_{exemplar}\right)+r\left(y_{i},m_{exemplar}\right)\\ & \;\;\;=argmax\left\{a\left(y_{i},y_{j}\right)+r\left(y_{i},y_{j}\right)\right\}\}. \end{array}$$

From above, the AP process with respect to the candidate node is conducted as follows:


Step 1: For ${\dot{y}}_{i},{\dot{y}}_{j}\in {\dot{Y}}_{candidate\;node}$, initialize the responsibility value $r\left({\dot{y}}_{i},{\dot{y}}_{j}\right)$ as the Euclidean distance $d\left({\dot{y}}_{i},{\dot{y}}_{j}\right)$ with Equation (33), and set the availability value $a\left({\dot{y}}_{i},{\dot{y}}_{j}\right)$ to zero;Step 2: Update the responsibility value $r\left({\dot{y}}_{i},{\dot{y}}_{j}\right)$ with Equation (34);Step 3: Update the availability value $a\left({\dot{y}}_{i},{\dot{y}}_{j}\right)$ with Equations (35) and (36);Step 4: If $r\left({\dot{y}}_{i},{\dot{y}}_{j}\right)$ and $a\left({\dot{y}}_{i},{\dot{y}}_{j}\right)$ become stable, conduct Step 5, otherwise return to Step 2;Step 5: For each sample ${\dot{y}}_{i}\in {\dot{Y}}_{candidate\;node}$, assign the sample data corresponding to $a\left(y_{i},y_{j}\right)+r\left(y_{i},y_{j}\right)$ as exemplar node $m_{exemplar}$ and then generate the exemplar set $M_{exemplar}$;Step 6: Separate $S_{father}$ with Equation (37).


### 3.3. Generate the Fault Diagnostic Tree under Minimum Conditional Criterion

Based on soft-sensor construction and subset division, the fault states of the target system can be located by cutting the set of potential failure sets with the function of sequences of soft-sensor nodes. In this section, we introduce how to generate the fault tree using a potential soft-sensor node under heuristic strategy based on the minimum conditional criterion. For the assigned failure set $S_{father}$, the contains numbers of potential sensor nodes corresponding to the candidate procedures. To choose the soft sensor for fault tree construction, the condition entropy $H(Y_{node}|{\dot{Y}}_{node},ELM_{candidate})$ is introduced as follows:(38)$$H\left(Y_{node}|{\dot{Y}}_{node},ELM_{candidate}\right)=-\sum_{y\in Y_{node}}p\left(y,\dot{y}|ELM_{candidate}\right)log(y|\dot{y,}ELM_{candidate})\;\;$$

Since the soft-sensor model is data-driven, the estimation value of the conditional entropy is computed as follows:(39)$$H\left(Y_{node}|{\dot{Y}}_{node},ELM_{candidate}\right)=\sum_{y\in Y_{node}}logp(y|\dot{y,}ELM_{candidate})$$

Assuming the data information $\left(Y_{node},{\dot{Y}}_{node}\right)$ is subject to Gaussian distribution, then $\dot{H}\left(Y_{node}|{\dot{Y}}_{node},ELM_{candidate}\right)$ is simplified as follows:(40)$$\dot{H}\left(Y_{node}|{\dot{Y}}_{node},ELM_{candidate}\right)=\sum_{y\in Y_{node}}{\left|\left|Y_{node}-{\dot{Y}}_{node}\right|\right|}^{2}$$

For all candidate soft-sensor nodes, the node with the lowest conditional entropy is selected to build the fault diagnostic tree. From above, the process to construct the fault diagnostic tree follows:

Step 1.Initialize the root node $Node_{root}$ of the decision tree. Assign the father fault set $S_{father}$ as the total fault set $S_{total}$ and set the data set $X_{node}$ as the whole training data set. Take all the test procedures as the potential test set $T_{potential}$ for $Node_{root}$. Set $T_{sequence}^{*}$ to ∅.Step 2.Generate the potential soft-sensor node $Node_{candi,t_{potential}}$ for each test procedure $t_{potential}\in T_{potential}$ based on Equations (24), (28) and (31).Step 3.Evaluate with Equation (39) the condition entropy $\dot{H}\left(Y_{node}|{\dot{Y}}_{node},ELM_{candidate}\right)$ for the entire potential soft-sensor node. Select the corresponding soft-sensor node $Node_{candi,t_{potential}}$ with the lowest condition entropy as the target node to the diagnostic tree, and update the $T_{sequence}^{*}$ as $\left\{T_{sequence}^{*},t_{opt}\right\}$. Additionally, remove $t_{opt}$ from $T_{potential}$.Step 4.Apply AP to separate the fault set $S_{father}$ based on ${\dot{Y}}_{node}$ and assign the exemplar reference $M_{exemplar}$ with Equation (36).Step 5.For each subset $S_{i}$ in $S_{son}$, construct the extending node $Node_{S_{i}}$. For each extending node $Node_{S_{i}}$, the father set is assigned as $S_{i}$ and the data set is constructed based on AP result. $T_{sequence}^{*}$ is initialized as $\left\{T_{sequence}^{*},t_{opt}\right\}$.Step 6.Generate the subset node by repeating **Steps 2** to **5** until reaching the minimal subset of failure mode. The construction process is completed when all the subnodes of the failure tree are constructed.

With the minimum conditional criterion, the fault diagnostic tree is generated with data-driven mechanisms and requires few engineering experiences. Thus, the generating process is applicable to complex systems with insufficient knowledge about structures, functions, and mechanisms.

With the diagnostic tree generated, the diagnostic process of the target system is implemented as follows:

Step 1.Initialize the target node $Node_{target}$ with the root node $Node_{root}$ of the diagnostic tree. Assign the potential fault set $S_{potential}$ to the total fault set $S_{total}$. Set the target sensor recording $x_{target}$ as ∅ and set the test sequence $T_{sequence}$ to ∅.Step 2.Conduct the new test procedure $t^{*}$ in the $Node_{target}$ and obtain the sensor recording $x_{target,t^{*}}$. Add $t^{*}$ into $T_{sequence}$ and merge $x_{target,t^{*}}$ into $x_{target}$. Compute the estimation of the target system ${\dot{y}}_{target}$ with $ELM_{node}$ in $Node_{target}$ by Equations (29) and (30).Step 3.Find the optimal $m_{exemplar}$ in $M_{exemplar}$ of $Node_{target}$ with smallest distance. Locate the subfault set $S^{*}$ as the updated $S_{potential}$ with Equation (37).Step 4.Search the soft-sensor node in the diagnostic tree with $S_{father}=S_{potential}$ and assign the corresponding node as the new target node $Node_{target}$.Step 5.Continue the diagnostic procedure by steps 2 to 4 until reaching the minimal subset of the failure set. After the diagnosis is finished, achieve the final estimation by using Equation (17).

## 4. Experiment

In this section, we use the analog circuit in [40] to evaluate the detection performance of the proposed method with state-of-art methods. As Figure 5 shows, the target system contains four second-order filters and one adding device. The detail of the system is presented in Table 1. The tolerance of R_1_, R_2_, R_3_, R_4_, R_5_, R_6_, R_7_, and R_8_ is ±10% while the tolerance of R_9_, R_10_, and R_11_ is ±1%. For capacitances, the tolerance is set to ±5%. Under healthy operation, the transmission gain of Av_1_, Av_2_, Av_3_, and Av_4_ is within a range of ±1%.

Here, the failures caused by different changes of amplifiers are taken into failure detection. The failure modes are defined based on the range of transmission gain for Av_1_, Av_2_, Av_3_, and Av_4,_ as shown in Table 2. Since 80% of failures in real applications have a single-failure mode, we only consider failure detection of single failure modes. For example, the failure condition of Av1 is divided into five phases with different ranges of transmission gain while the transmission gain of Av_2_, Av_3_, and Av_4_, are collected with four different frequencies (10 Hz, 100 Hz, 10 kHz, and 100 kHz) of input signals. The details are shown in Table 3. These voltage outputs from Av_1_, Av_2_, Av_3_, and Av_4_ are regarded as the potential test points for failure detection. In total, there are 16 candidate test points and 17 potential fault states that consider the health state.

We apply Monte Carlo simulation to generate 20 samples for each failure mode by using Pspice software to access data.

According to the traditional PHM framework shown in Figure 3, the optimization process requires binary fault marks based on human experience and design detection circuits for each fault state. Since there are four fault states for each second-order filter, the detection is a large burden on circuit design. Testability design may also fail to relate the relationship between binary estimations of different test procedures. On the other hand, the maintenance design suffers low estimation efficiency as the testability does not consider the detailed information of sensor recordings.

Unlike the sequential framework, the proposed method considers the direct sensor information under testability and maintenance design. With cooperative procedures in Figure 4, the proposed method generates the candidate soft-sensor nodes for maintenance. At the same time, the testability design uses the minimal conditional criterion to generate the optimized diagnostic strategy. The minimal conditional criterion enhances the flexibility of testability design by considering the soft-sensor estimation in maintenance phases. On the other hand, the maintenance performance with full sensor recording increases information usage efficiency. Instead of human-experience processing, the proposed method saves many costs during the PHM design. To evaluate diagnostic performance, we compared our method with the hidden Markov method (HMM), support learning machine (SVM), and radial basis function (RBF) by using all recordings of 16 test points as input information. To estimate the feature extraction performance and learning machine function, we also took HMM and SVM with principal component analysis (PCA) and extreme learning machine (ELM) into comparison. For each method, we used 70% of the samples in each fault state as the training samples to construct the model and the other 30% as the sensor information of target systems. We assigned 100 kernels or hidden nodes for our soft-sensor nodes for the SVM, RBF, and ELM models. Each method was conducted 30 times to obtain the average performance.

Table 3 shows the FAR, FDR, and accuracy of each method. HMM has high diagnostic accuracy without feature extraction, especially for S_0_. This is because HMM has a higher statistical analysis ability than SVM and RBF. However, HMM gives poorer FAR than the other methods. Unlike HMM and RBF, SVM and ELM have lower FAR and become more sensitive to false-negative samples by the advantage of the learning machine. Comparing HMM, SVM, PCA–HMM, and PCA–SVM, PCA improves HMM diagnosis for S5 and S8 and proves that proper feature extractions can benefit from the diagnostic performance. Our method has the lowest FAR, the highest FDR, and the highest accuracy compared with the other methods. Based on the same sensor recordings or even less information, the generated strategy provides an accurate location for all test samples for all 16 fault states. Hence, with ensemble learning based on soft sensors, the functions of sensor recording are largely improved. 

Figure 6 shows the diagnostic tree of our method. All fault conditions are separated and recognized with 13 individual testing sequences with the tree structure. Each testing sequence takes less than 5 test procedures and the whole diagnostic tree contains only 9 testing points out of 16 potential test points. In other words, the fault state of the target systems is located within 2–5 test procedures instead of collecting all 16 sensor recordings. From above, the diagnostic tree has higher efficiency and lower cost testing consumptions than diagnostic strategies that require full-tests built by SVM, HMM, and ELM, as well as the constrained diagnostic tests strategies that require PCA-based methods. Additionally, unlike traditional diagnostic trees with binary structures, our method separates the fuzzy set with the evaluation results from constructing and dividing modules. As a result, our testability design extends the diagnostic flexibility and improves the diagnostic accuracies of each fault mode.

Based on soft-sensor construction, the potential test accuracies of the diagnostic tree root node are compared based on different test procedures. As shown in Figure 7, different test procedures differ on diagnostic accuracies, especially for S_0_–S_5_. Therefore, the minimum conditional criterion-based fault tree constructions can efficiently select proper soft-sensor nodes into the diagnostic tree and ensure that the diagnostic FAR and FDR and their accuracies receive high improvement under the constructing process.

The affinity propagation result from the diagnostic-tree-root node is presented in Figure 8. As the root nodes evaluate all samples, the affinity results are generated with fault mode evaluations of all 17 fault conditions (from S_0_ to S_16_). Here, we depict the affinity propagation result from three fault states, which occur at the same space, such as Av_1_ failures (S_0_, S_1_, and S_2_), Av_2_ failures (S_5_, S_7_, and S_14_), and Av4 failures (S_12_, S_13_, and S_15_). Additionally, we present the affinity propagation results with three failure modes in different places: S_2_ in Av_1_, S_7_ in Av_2_, and S_14_ in Av_4_. 

From Figure 8, the soft-sensor nodes provide distinguishable BPA evaluations for each data point. The affinity propagation generates sample clusters from different dimensions adaptively with the data point similarity measurements. Most data points are clustered with topological closer clusters; others may differ on other dimensions. Compared with traditional clustering strategies for diagnosis, affinity propagation provides practical, automatic fault-state divisions and saves much human labor on engineering applications. 

Finally, the sequence testing performance for fault states is shown in Figure 9. Here, we present the test sequence of S_0_, S_3_, S_8_, and S_15_. These four test sequences achieve their best diagnostic accuracy within three to five test procedures, and the test efficiency is much higher than traditional maintenance methods. Test accuracy grows increasingly for all test sequences as test nodes are added into the test procedures. Especially for the test sequence of S_0_, the diagnostic accuracy is less than 60% in the first test procedures as the corresponding fuzzy state set contains many members. However, the accuracy grows fast as more nodes are added into the sequences and the potential states become smaller. Thus, the ensemble function of soft-sensor nodes improved the diagnostic performance with high efficiency.

From above, our method has better diagnostic accuracies and lower FARs compared with other state-of-art diagnostic methods. Additionally, our diagnostic strategies take only 9 out of 16 tests points and save much test assignment consumption. For each fault state, the diagnostic procedures provide 1 test sequence within 5 test procedures. Thus, the diagnostic procedure enjoys high efficiency for applications. Finally, the affinity propagation enlarges the diagnostic flexibility and saves much human labor on testability design.

## 5. Conclusions

Along with a short production cycle and rapid development of design technology, existing PHM techniques have become impractical and fail to match the structural and functional complexity. Prior knowledge preparation costs too much in human labor and binary decision-making strategies waste the entailed sensor recording, especially for large complicated systems. 

We propose a test strategy optimization based on soft sensing and ensemble belief measurement to overcome these weaknesses. The proposed method constructs a closed loop between testability design and maintenance design, generating an efficient fault diagnostic tree with ELM-based soft-sensor nodes. Unlike traditional diagnostic approaches, our diagnostic tree adaptively separates the fault sets by affinity propagation, and the soft-sensor nodes are assigned with the minimum conditional criterion. Thus, our methods can achieve high efficiency and flexibility for diagnostic processes.

The experiment results prove that our methods have minimum FAR and maximum accuracies on fault diagnosis among state-of-art methods. Additionally, our methods require fewer test procedures and increase the test efficiency compared with other methods. Because the construction processes are based on ELM and AP, the PHM design saves much human labor and becomes more flexible compared to traditional PHM approaches. Hence, the proposed method has good performance on test strategy design. However, the proposed method uses an offline construction technique for the diagnostic tree. As a result, the diagnostic performance only depends on the assigned fault set, and the recordings of online operations do not work on the PHM design. Therefore, the online updating of the diagnostic strategy should be further investigated.

## Figures and Tables

**Figure 1 sensors-22-02138-f001:**
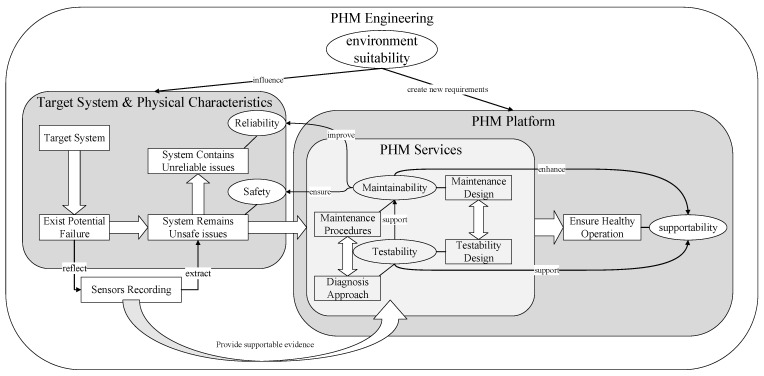
Element and the corresponding relationship of PHM engineering.

**Figure 2 sensors-22-02138-f002:**
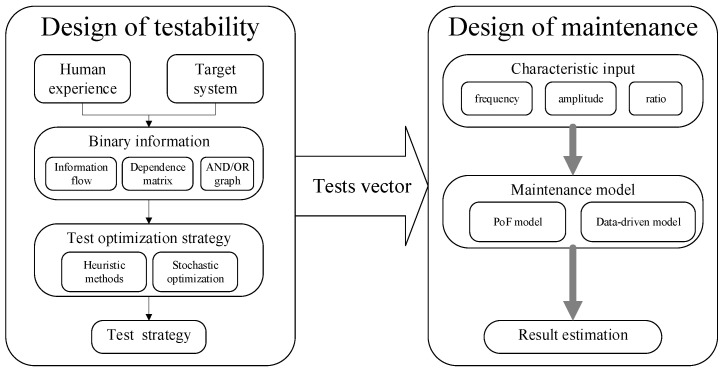
Traditional framework of testability and maintenance design.

**Figure 3 sensors-22-02138-f003:**
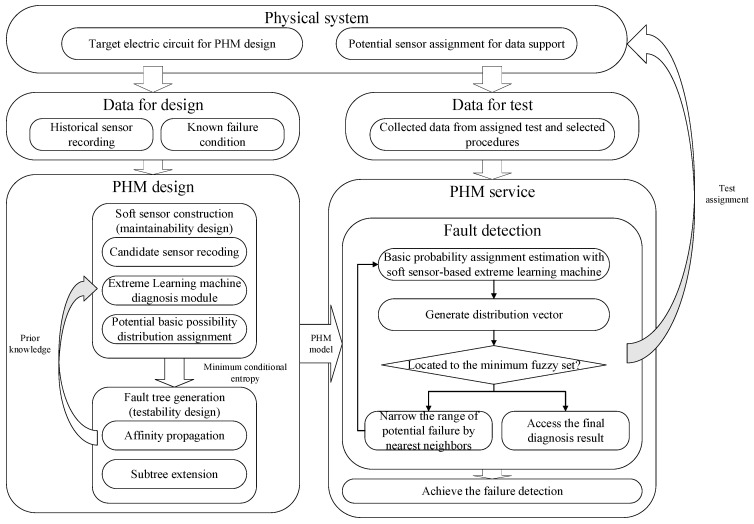
General framework of test strategy optimization based on soft sensing and ensemble belief measurement.

**Figure 4 sensors-22-02138-f004:**
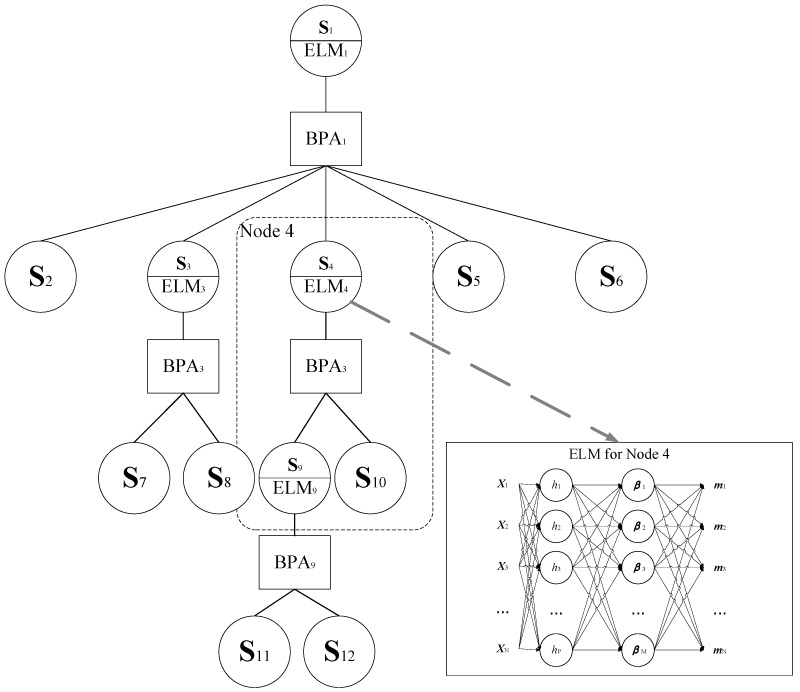
Fault tree of test strategy optimization based on soft sensing and ensemble belief measurement.

**Figure 5 sensors-22-02138-f005:**
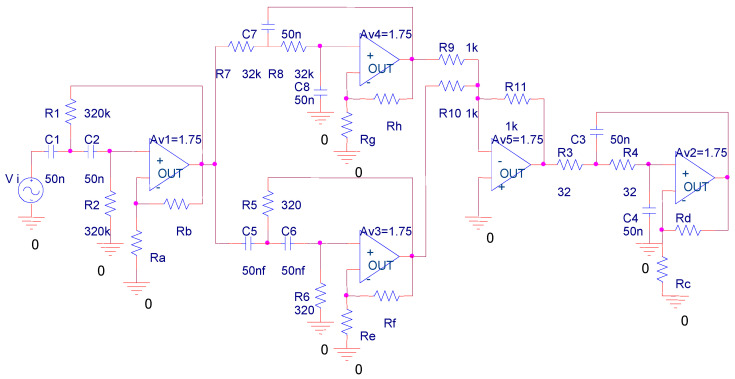
Target analog circuit.

**Figure 6 sensors-22-02138-f006:**
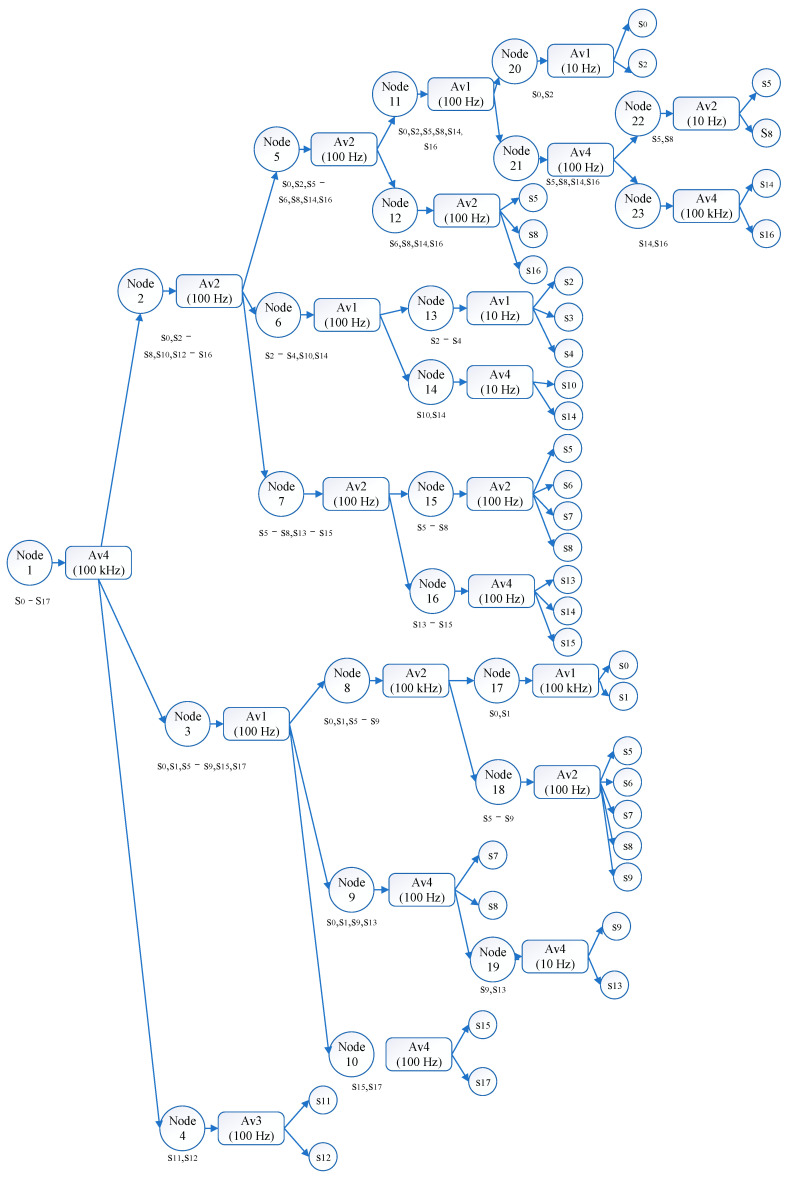
Analog circuit diagnostic tree.

**Figure 7 sensors-22-02138-f007:**
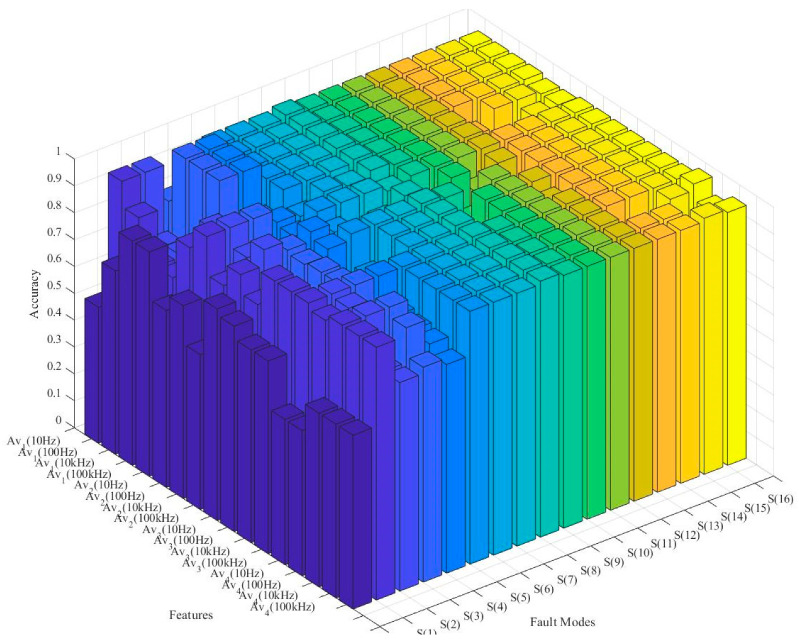
Analog circuit diagnostic tree potential ELM model accuracy comparison.

**Figure 8 sensors-22-02138-f008:**
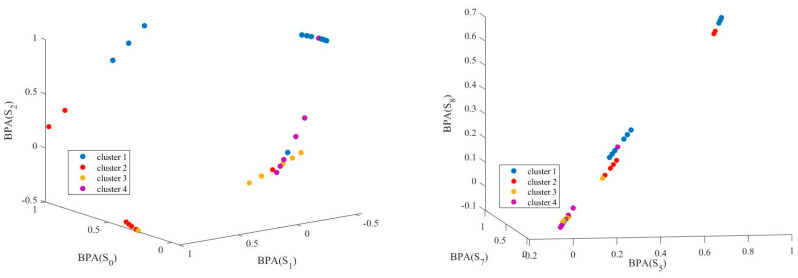
Potential ELM model accuracy comparison.

**Figure 9 sensors-22-02138-f009:**
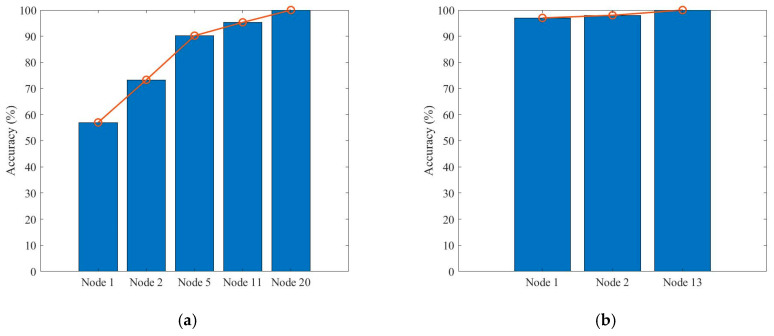
Test sequence accuracy comparison: (**a**) S_0_ test sequence, (**b**) S_3_ test sequence, (**c**) S_8_ test sequence, and (**d**) S_15_ test sequence.

**Table 1 sensors-22-02138-t001:** Details for the components of the circuit.

Components	Nominal Value	Tolerance	Subsystem
R_1_	320 kΩ	10%	High-Pass Filter 1 F_1_ = 10 Hz
R_2_	320 kΩ	10%	
C_1_	50 nF	5%	
C_2_	50 nF	5%	
Av_1_	1.75	1%	
R_3_	32 Ω	10%	Low-Pass Filter 1 F_2_ = 100 kHz
R_4_	32 Ω	10%	
C_3_	50 nF	5%	
C_4_	50 nF	5%	
Av_2_	1.75	1%	
R_5_	320 Ω	10%	High-Pass Filter 2 F_3_ = 10 kHz
R_6_	320 Ω	10%	
C_5_	50 nF	5%	
C_6_	50 nF	5%	
Av_3_	1.75	1%	
R_7_	32 kΩ	10%	Low-Pass Filter 2 F_4_ = 100 Hz
R_8_	32 kΩ	10%	
C_7_	50 nF	5%	
C_8_	50 nF	5%	
Av_4_	1.75	1%	
R_9_	1 kΩ	1%	Adder
R_10_	1 kΩ	1%	
R_11_	1 kΩ	1%	

**Table 2 sensors-22-02138-t002:** Denotation of fault states.

Fault Index	Av_1_ Value Range	Av_2_ Value Range	Av_3_ Value Range	Av_4_ Value Range
S_0_(normal)	(1.70,1.80)	(1.70,1.80)	(1.70,1.80)	(1.70,1.80)
S_1_	(1.60,1.70)	(1.70,1.80)	(1.70,1.80)	(1.70,1.80)
S_2_	(1.80,1.90)	(1.70,1.80)	(1.70,1.80)	(1.70,1.80)
S_3_	(1.50,1.60)	(1.70,1.80)	(1.70,1.80)	(1.70,1.80)
S_4_	(1.90,2.00)	(1.70,1.80)	(1.70,1.80)	(1.70,1.80)
S_5_	(1.70,1.80)	(1.60,1.70)	(1.70,1.80)	(1.70,1.80)
S_6_	(1.70,1.80)	**(1.80,1.90)**	(1.70,1.80)	(1.70,1.80)
S_7_	(1.70,1.80)	(1.50,1.60)	(1.70,1.80)	(1.70,1.80)
S_8_	(1.70,1.80)	(1.90,2.00)	**(1.70,1.80)**	(1.70,1.80)
S_9_	(1.70,1.80)	(1.70,1.80)	(1.60,1.70)	(1.70,1.80)
S_10_	(1.70,1.80)	(1.70,1.80)	(1.80,1.90)	(1.70,1.80)
S_11_	(1.70,1.80)	(1.70,1.80)	(1.50,1.60)	(1.70,1.80)
S_12_	(1.70,1.80)	(1.70,1.80)	(1.90,2.00)	(1.70,1.80)
S_13_	(1.70,1.80)	(1.70,1.80)	(1.70,1.80)	(1.60,1.70)
S_14_	(1.70,1.80)	(1.70,1.80)	(1.70,1.80)	(1.80,1.90)
S_15_	(1.70,1.80)	(1.70,1.80)	(1.70,1.80)	(1.50,1.60)
S_16_	(1.70,1.80)	(1.70,1.80)	(1.70,1.80)	**(1.90,2.00)**

**Table 3 sensors-22-02138-t003:** Performance comparison.

Method	Performance	S0	S1	S2	S3	S4	S5	S6	S7	S8	S9	S10	S11	S12	S13	S14	S15	S16
HMM	FAR	25.0	0.00	0.00	0.00	0.00	0.00	0.00	20.00	33.3	0.00	20.0	0.00	0.00	0.00	52.6	33.3	16.7
	FDR	96.3	100	100	100	100	98.8	96.3	100	100	100	100	100	100	98.7	98.7	100	98.7
	accuracy	95.3	100	100	100	100	97.7	95.3	100	98.9	100	100	100	100	95.3	96.5	98.8	98.8
SVM	FAR	0.00	0.00	0.00	0.00	0.00	0.00	0.00	0.00	0.00	0.00	0.00	0.00	0.00	0.00	0.00	0.00	0.00
	FDR	92.3	94.1	94.1	98.8	96.4	94.1	94.1	97.6	100	94.1	94.1	94.1	94.1	94.1	94.1	96.4	97.6
	accuracy	92.9	94.1	94.1	98.9	96.5	94.1	94.1	97.7	100	94.1	94.1	94.1	94.1	94.1	94.1	96.5	97.7
RBF	FAR	33.3	66.7	33.3	33.2	36.5	0.00	0.00	37.5	0.00	0.00	0.00	50.0	60.0	33.3	60.0	54.4	28.6
	FDR	92.9	95.1	93.9	98.7	100	96.4	100	100	100	95.2	94.1	100	98.7	96.3	96.3	100	100
	accuracy	91.7	93.0	95.1	96.5	90.6	96.5	96.5	96.5	100	96.5	100	95.3	94.1	94.1	95.3	93.0	97.7
PCA HMM	FAR	25.0	0.00	0.00	0.00	0.00	0.00	20.0	0.00	0.00	0.00	0.00	0.00	0.00	0.00	28.6	0.00	0.00
	FDR	96.3	100	100	100	100	100	98.8	100	100	100	100	100	100	100	100	100	100
	accuracy	95.3	100	100	100	100	100	97.7	100	100	100	100	100	100	100	97.7	100	100
PCA SVM	FAR	0.00	0.00	0.00	0.00	0.00	0.00	2.30	0.00	0.00	0.00	0.00	0.00	0.00	0.00	0.00	0.00	0.00
	FDR	93.0	94.1	94.1	98.8	96.4	94.1	94.1	97.6	100	94.1	94.1	94.1	94.1	94.1	94.1	96.4	97.6
	accuracy	93.0	94.1	94.1	98.9	96.5	94.1	97.7	100	94.1	94.1	94.1	94.1	94.1	94.1	94.1	96.5	97.7
ELM	FAR	0.00	0.00	0.00	0.00	0.00	0.00	2.30	0.00	0.00	0.00	0.00	0.00	0.00	0.00	0.00	0.00	0.00
	FDR	91.4	100	100	100	100	100	100	100	100	100	100	100	100	100	100	96.4	97.6
	accuracy	92.3	100	100	100	100	100	100	100	100	100	100	100	100	100	100	96.4	97.6
OURS	FAR	0.00	0.00	0.00	0.00	0.00	0.00	0.00	0.00	0.00	0.00	0.00	0.00	0.00	0.00	0.00	0.00	0.00
	FDR	99.7	100	100	100	100	100	100	100	100	100	100	100	100	100	100	100	100
	accuracy	100	100	100	100	100	100	100	100	100	100	100	100	100	100	100	100	100

## Data Availability

The data that support the findings of this study are available on request from the authors.

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
