# Peer review of "Test Strategy Optimization Based on Soft Sensing and Ensemble Belief Measurement"

_sensors, 2022, doi:10.3390/s22062138_

Round 1

Reviewer 1 Report

Testing optimization is always topical and plays a vital role when systems become complex.

The authors introduce a new fresh approach.

The writing style of the article is clear and straightforward. There is always some article like challenges. The background and details are well presented. Visuality has been used appropriately.

Author Response

Thank you very much for your appreciation.

Reviewer 2 Report

The current paper proposes a test strategy optimization based on soft-sensing and ensemble belief measurement to overcome these weaknesses. The proposed method constructs a close loop between testability design and maintenance design and generating an efficient fault diagnosis tree with ELM-based soft sensor nodes. The theory is validated using simulations.

Comments to authors:

- Please add more details regarding paper’s novelty, it is not very clear what are the novelties of this paper.

- Please add more details of how the theory from the first sections is applied in the results section.

- The authors can add the steps of implementing the algorithms. The theoretical part can be better detailed. The steps will be in the benefit of the readers, maybe they’ll help the readers to implement the proposed algorithm.

- Please define the objective function, since the paper is related to optimization.

- The state of the art it is very poor regarding representative papers, maybe the author could add the following publications:

o Hybrid Data-Driven Fuzzy Active Disturbance Rejection Control for Tower Crane Systems, European Journal of Control, vol. 58, pp. 373-387-11, 2021.

o Event-Triggered Adaptive Fuzzy Control for Stochastic Nonlinear Systems with Unmeasured States and Unknown Backlash-Like Hysteresis, IEEE Transactions on Fuzzy Systems, doi 10.1109/TFUZZ.2020.2973950, pp. 1–19, 2020.

- Please add more details regarding the obtained results.

- Add the both the advantages and the disadvantages of the proposed method. In the proposed manuscript only the advantages are presented.

Reviewer 3 Report

The subject of the paper represents a good contribution related to improve a test strategy optimization based on soft-sensing and ensemble belief measurements to overcome the problems generated by improper testability design.

The paper is well structured and presents an interest from scientific and technological point of view. The analyzed paper is characterized by a suitable and adequate title, and a clear and pertinent exposure. Their goal is clearly written. The English is clear that makes the paper very easy to understand.

Please ensure that citations within the text are in the correct format; the references at the end of the text are in the correct format; figures are placed at appropriate positions within the text and are of suitable quality.

               Specific problems are:

1. Some key parameters related to the simulations and computational method should be mentioned.

2. Please, attention to the importance of citation from recent publication relating to your paper. Update the relevant literatures with special attention to papers published in 2020-2022.

Round 2

Reviewer 2 Report

The paper has been seriously improved since last revision. The authors answered to all my concerns. From my point of view the paper can be accepted to be published in Sensors journal.